# Sensor Fusion-Based Anthropomorphic Control of a Robotic Arm

**DOI:** 10.3390/bioengineering10111243

**Published:** 2023-10-24

**Authors:** Furong Chen, Feilong Wang, Yanling Dong, Qi Yong, Xiaolong Yang, Long Zheng, Yi Gao, Hang Su

**Affiliations:** 1Department of Mechanical Engineering, College of Mechanical and Electrical Engineering, Changchun University of Science and Technology, Changchun 130012, China; chenfurong0721@163.com (F.C.); feilong_wang0707@163.com (F.W.);; 2Key Laboratory of Bionic Engineering, Ministry of Education, Jilin University, Changchun 130022, China; 3Weihai Institute for Bionics, Jilin University, Weihai 264402, China; 4School of Foreign Languages & Literature, Shandong University, Jinan 250000, China; yanling.dong2@gmail.com; 5ESIEE Paris, 2 Boulevard Blaise Pascal, 93160 Noisy-le-Grand, France; qi.yong@edu.esiee.fr

**Keywords:** sensor fusion, Kalman filter, anthropomorphic control, joint angle snapping, robotic arm

## Abstract

The main goal of this research is to develop a highly advanced anthropomorphic control system utilizing multiple sensor technologies to achieve precise control of a robotic arm. Combining Kinect and IMU sensors, together with a data glove, we aim to create a multimodal sensor system for capturing rich information of human upper body movements. Specifically, the four angles of upper limb joints are collected using the Kinect sensor and IMU sensor. In order to improve the accuracy and stability of motion tracking, we use the Kalman filter method to fuse the Kinect and IMU data. In addition, we introduce data glove technology to collect the angle information of the wrist and fingers in seven different directions. The integration and fusion of multiple sensors provides us with full control over the robotic arm, giving it flexibility with 11 degrees of freedom. We successfully achieved a variety of anthropomorphic movements, including shoulder flexion, abduction, rotation, elbow flexion, and fine movements of the wrist and fingers. Most importantly, our experimental results demonstrate that the anthropomorphic control system we developed is highly accurate, real-time, and operable. In summary, the contribution of this study lies in the creation of a multimodal sensor system capable of capturing and precisely controlling human upper limb movements, which provides a solid foundation for the future development of anthropomorphic control technologies. This technology has a wide range of application prospects and can be used for rehabilitation in the medical field, robot collaboration in industrial automation, and immersive experience in virtual reality environments.

## 1. Introduction

Anthropomorphic technology has gained significant attention in the past few decades, particularly in the field of robotics [1]. The concept of anthropomorphism involves designing technology that mimics human-like characteristics, such as movement, dexterity, and intelligence, to perform various tasks [2]. Anthropomorphic control of robotic arms, in which the motion pose of a human arm is mapped onto a robotic arm, enabling the robotic arm to perform delicate tasks like a human arm, has become one of the most compelling applications in this field [3]. However, it is not easy to achieve anthropomorphic control. This control method usually requires the use of advanced sensors, motion control algorithms, and artificial intelligence technology to enable the robotic arm to autonomously perceive the environment, identify objects, plan motion strategies, and complete complex operational tasks [4]. The realization of anthropomorphic control can improve the operation accuracy, efficiency, and adaptability of the robotic arm and expand its application range, such as in the fields of manufacturing, medical care, and special environment detection [5].

The anthropomorphic control of robotic arms has achieved many research results and has been widely used in biomedical and industrial fields [6,7]. In 2010, Panagiotis K. Artemiadis et al. used human upper limb motion estimation to control anthropomorphic robot arms in 3D space in real time to help the elderly and disabled in their daily lives [8]. In 2017, Alkinoos Athanasiou et al. designed and developed off-the-shelf BCI-controlled anthropomorphic robotic arms for assistive technology and rehabilitation applications [9]. In 2018, José E. Naranjo et al. designed a bilateral remote operating system that allows operators to perform maintenance or remote inspection activities in a well site located in Petroamazonas EP, Ecuador [10]. In 2022, Valentina A. Yurova et al. developed an anthropomorphic multi-fingered artificial hand for robotics research and teaching applications [11].

Remote-controlled manipulators have attracted a wide range of research interest in recent years, with applications in a variety of fields such as medical rehabilitation, industrial automation, hazardous environment operations, and virtual reality. These systems aim to achieve teleoperation or anthropomorphic control through the use of sensors and control techniques that precisely transmit the operator’s movements and actions to the manipulator. Lorenzo Scalera et al. allowed the user to control the robot’s movements with their gaze through an eye-tracking system [12]. Lingyun Chen et al. proposed the use of digital tablets and motion capture suits to operate a tele-robot to draw portraits with smooth and even strokes [13].

Humanoid robots are also the most suitable platform for anthropomorphic control. The most common ones are full-body humanoid robots, such as WALK-MAN [14], upper-limb humanoid robots, and lower-limb humanoid robots. In contrast, upper-limb humanoid robots are more flexible and versatile in many applications. For example, flexible robotic arms mounted on drones can be used for aerial exchange tasks [15]. Therefore, we mounted an upper-limb robot on a mobile chassis to achieve anthropomorphic control.

Teleoperation technology plays an important role in enabling anthropomorphic control, combining the advantages of robots (with high levels of accuracy, speed, and repeatability) with the flexibility and cognitive skills of human workers to achieve efficient human–machine collaboration [16]. In recent years, A. Noccaro et al. proposed a redundant robotic arm remote control method based on IMU [17]. Julio C. Cerón et al. developed a remote operating system with an assistive robot (NAO) using Kinect V2, Meta Quest glasses, and a Nintendo Switch controller [18]. These studies emphasize the advantages of teleoperation technology in improving robot accuracy and anthropomorphism. Perception technology is also essential during teleoperation. Multimodal sensor systems provide operators with richer sensory information, especially tactile feedback [19], allowing them to better understand the robot’s status and the surrounding environment [20]. More often, it is used in the field of medical rehabilitation, such as the natural touch produced by prosthetic hand sensors stimulating nerves [21]. We use tactile sensors distributed on the dexterous finger pads of the robot to help sense external pressure. Although these research results are exciting, there are still some challenges in achieving a true level of anthropomorphism [22,23]. One of the most important challenges in realizing anthropomorphic control is to sense human motion information and map it to the robot’s decision-making [24,25]. The human arm has a complex biomechanical structure, including multiple degrees of freedom and flexible joints, which enables humans to perform various complex tasks [26,27]. In contrast, most robotic arms typically have fewer degrees of freedom and more rigid structures, thus requiring precise kinematic mapping for anthropomorphic control [28,29]. Solving this problem is crucial to improving the adaptability of robotic arms in various applications.

To achieve the smooth estimation of arm pose, sensor technology and advanced algorithms are being actively developed [30,31]. Vision-based sensors can detect multiple joints of the arm at the same time, achieve pose measurement, and track the movement trajectory of the arm, such as Leap Motion Controllers sensors and Microsoft Kinect sensors [32,33]. Myoelectric-based sensors, such as joint goniometers, inertial sensors, and electromyography, can monitor muscle contraction responses in real time and detect arm muscle activity levels in static and dynamic states and are accurate, fast, stable, and adaptable [34]. However, it is difficult to build a complete arm kinematics model with only one modal sensor [35]. Because the movements of the hands are often subtle, it is difficult for vision sensors to capture wrist and finger movements while capturing the shoulder and elbow, while myoelectric sensors can easily capture them. Visual sensors capture the shoulder and elbow more directly than myoelectric sensors [36], and the algorithms are relatively mature and advanced [37,38]. The fusion of multiple sensors to capture arm movements not only establishes kinematic models of shoulders, elbows, wrists, and hands [39] but also improves the accuracy and robustness of the system and is applicable to and expands complex application scenarios [40]. In 2018, Yi-Chun Du et al. used IMU to calibrate Kinect’s skeletal tracking for upper limb rehabilitation games, though they did not apply it to the anthropomorphic control of robotic arms [41]. In 2020, Yangyang Zhang et al. used wearable data belts and data gloves to accurately measure the joint motion of shoulder joints, elbow joints, wrist joints, metacarpal bones, and proximal finger joints and experimentally verified the effectiveness of the proposed method [42]. More, we have conducted related work to integrate Kinect with data gloves [43].

In this paper, we propose the use of multimodal sensors to map the kinematics of the arm to a robotic arm for anthropomorphic control. The following are the research contributions of this paper.

(1)A variety of sensor technologies, including Kinect, IMU, and a data glove, are integrated to build a comprehensive arm motion model covering key joints such as the shoulder, elbow, wrist, and hand. This comprehensive model provides a solid foundation for the fine-grained control of robotic arms, allowing robots to more naturally mimic human arm movements.(2)The Kalman filter fusion technology is adopted to fuse the data from the Kinect and IMU sensors together, which effectively overcomes the problems of sensor errors and noise. High-precision estimation of the joint angles of the manipulator can be obtained, which ensures the stability and accuracy of the robot in various tasks.(3)Through the anthropomorphic control technology in this study, various anthropomorphic actions of the manipulator are realized. This includes movements such as shoulder abduction, flexion, and rotation, elbow flexion and extension, wrist rotation, and flexion of the fingers, allowing the arm to adapt to different anthropomorphic tasks, including grasping.

## 2. System Description

In this paper, an upper limb skeletal tracking system was developed using multi-sensors and applied to control a self-developed robotic arm. The general setup of the proposed operational control system is shown in Figure 1. The operator wears a haptic device consisting of the system of three IMUs and a data glove, which are placed on the experimenter’s right arm. At the same time, the system is equipped with a Kinect vision device (Microsoft Corporation, Redmond, WA, USA), whose optimal distance measurement range is 1.2 m to 3.5 m. One of the core functions of the system is to fuse the motion data of the shoulder and elbow joints. The Kinect and IMU systems provided data from multiple sources, which we efficiently fused using Kalman filters to obtain highly accurate estimates of these joint angles. This fusion technique not only improves the accuracy of the system but also enhances the ability to adapt to dynamic environmental changes, thus ensuring the stable movement of the robotic arm. In addition to shoulder joints and elbow joints, we also collect data on wrist joints and finger joints through data gloves to control the manipulator. This meticulous data collection allows the robotic hand to mimic the movements of the human hand, giving it a high degree of dexterity and precision, especially when grasping movements are required.

This paper uses a previously developed robotic arm, coupled with a dexterous hand, as shown in Figure 2, as the execution part of the entire system. The robotic arm has five degrees of freedom, which are composed of two joint angles of the shoulder, two joint angles of the elbow, and one joint angle of the wrist. These five degrees of freedom are mapped and controlled by the human shoulder and elbow joint angles. They are shoulder yaw, shoulder pitch, shoulder roll, elbow pitch, and wrist roll, respectively. The manipulator has 6 degrees of freedom, which include two degrees of freedom for the thumb and one bending degree for each of the other four fingers. These degrees of freedom enable the robotic arms and hands to achieve highly flexible movements that can be adapted to the needs of many different tasks. In addition, the fingertips of the manipulator are also equipped with tactile sensors, which can achieve high-precision tactile perception.

As shown in Figure 1, the operating system mainly consists of three parts: a motion tracking end, robot remote end and communication protocol.

(1)The motion tracking end consists of the operator, three IMUs, data gloves, Kinect camera and remote host. It can collect the upper limb movement posture of the operator’s right arm, including the position and rotation of the shoulder, elbow, wrist and hand.(2)The remote end of the robot includes a robotic arm, a robotic hand, and a robot host. The robotic arm is driven by five servos, while the manipulator is driven by six pushrod motors. The five fingertips of the manipulator are equipped with tactile sensors that can output the pressure of the contact object. Two STM32 controllers are used as the control system, and expansion boards are used for assistance.(3)In terms of communication protocol, the remote host collects data from the motion tracking end. The IMU system uses Bluetooth BLE5.0, the data glove uses LAN to communicate through the receiver, and Kinect communicates through USB3.0. The remote host processes and fuses the data and publishes the upper limb joint angles to the ROS network, and the robot host subscribes to messages and issues instructions to the control system through TCP/IP. The tactile feedback of the manipulator feeds back the fingertip pressure to the remote host through UDP/IP.

The design and composition of the entire system enables us to achieve highly anthropomorphic control of the robotic arm, naturally mapping human motion postures to the robotic arm and providing a higher degree of automation and intelligence for various application scenarios. This system has broad application prospects in the fields of medical treatment, industry, and special environment detection, opening up new possibilities for the development of robotics.

## 3. Methodology

### 3.1. Angle Algorithm for Kinect

Kinect is a human behavior perception sensor, which obtains human skeleton point data through a depth camera and infrared sensor, so as to realize high-precision tracking of human movement. The core of this technology is to use the depth image to recognize the shape and posture of the human body and at the same time perform spatial modeling to obtain key skeletal point information in real-time operation. The depth image provides Kinect with the distance data between the surface of the human body and the sensor, enabling it to accurately capture the position of various key points of the human body. The ability of Kinect is not limited to the capture of skeletal points but also includes real-time tracking of human body movement, which enables us to capture the changes in joint angles when the operator performs various actions, providing important data support for subsequent robotic arm control. Through Kinect’s highly accurate angle calculation, we can achieve accurate tracking and control of the upper limb joint angle, thus laying a solid foundation for the anthropomorphic control of the robotic arm.

As shown in Figure 3, the shoulder (points A and B), elbow (point C), wrist (point D), hand (point E), and hip (point O) are bone points extracted by Kinect. According to the principle of vector approach by Reddivari et al. [44], we calculate the joint angle of the shoulder and elbow. Plane XY and plane YZ are the operator’s body level and body orientation, respectively. The vector $\vec{BF}$ and vector $\vec{BG}$ are the mapping of vector $\vec{BC}$ on plane YZ and plane XY, respectively, the angle ∠OBF is the shoulder yaw, and the angle ∠OBG is the shoulder pitch. Angle ∠DCP is the elbow pitch. The normal vectors of plane OBC and plane BCD are vector $\vec{BM}$ and vector $\vec{CN}$, respectively. Vector $\vec{BH}$ is parallel to vector $\vec{CN}$, and the angle ∠HBM is the shoulder roll. The formula used in the calculation process is as follows. The calculation formula of the vector is
(1)$$\vec{AB}=\left(x_{b},y_{b},z_{b}\right)-\left(x_{a},y_{a},z_{a}\right).$$

The known vector $\vec{BO}=\left(x_{1},y_{1},z_{1}\right)$ and vector $\vec{BC}=\left(x_{2},y_{2},z_{2}\right)$ calculate their normal vector as
(2)$$\begin{array}{cc} \vec{BM} & =\vec{BO}\times \vec{BC}\\ \; & =\left(y_{1}z_{2}-y_{2}z_{1},z_{1}x_{2}-x_{1}z_{2},x_{1}y_{2}-x_{2}y_{1}\right). \end{array}$$

The same logic can calculate the vector $\vec{CN}$. We compute the angle from vector $\vec{a}$ and vector $\vec{b}$ as
(3)$$\begin{array}{cc} \; & Cos\theta =\frac{\vec{a}\cdot \vec{b}}{\left|\vec{a}||\vec{b}\right|}\\ \; & \theta ={cos}^{-1}\left(Cos\theta \right) \end{array}$$

For example, the elbow pitch angle is calculated as follows:(4)$$\begin{array}{cc} \; & Cos∠DCP=\frac{\vec{BC}\cdot \vec{CD}}{\left|\vec{BC}||\vec{CD}\right|}\\ \; & ∠DCP={cos}^{-1}\left(Cos∠DCP\right). \end{array}$$

### 3.2. Angle Algorithm for IMU System

The IMU (inertial measurement unit) is a key component of our system. The IMU948 model by Chenyi Electronic Technology was selected, as shown in the Figure 4. The IMU parameter specifications used are shown in Table 1. They include a three-axis gyroscope, three-axis accelerometer, and three-axis magnetometer and support Bluetooth 5.0 wireless technology. This IMU is equipped with an improved Kalman fusion algorithm and an anti-magnetic interference fusion algorithm, which has many advantages, such as strong real-time performance, high precision, and good stability. These properties make the IMU system ideal for acquiring joint angles of the upper extremity.

In order to collect the angle of the upper limb joints of the operator, the data collected from the IMU system are mapped to the human kinematics model, as shown in Figure 5. The three IMUs placed on the chest, upper arm, and forearm of the human body collect real-time upper arm postures to study the joint angles of the shoulder joint and elbow joint. After the three IMUs are turned on, the operator’s arms hang vertically in the initial position, as shown in Figure 5. The purpose of initialization is to find the constant relationship between the parent bone and child bone, such as the body and arm and the arm and forearm.
(5)$$\begin{array}{cc} \; & {}^{Body}R_{Arm}^{const}={\left({}^{RG}R_{IMU0}^{init}\right)}^{-1}\cdot {}^{RG}R_{IMU1}^{init}\\ \; & {}^{Arm}R_{Forearm}^{const}={\left({}^{RG}R_{IMU1}^{init}\right)}^{-1}\cdot {}^{RG}R_{IMU2}^{init} \end{array}$$
where ${}^{Body}R_{Arm}^{const}$ and ${}^{Arm}R_{Forearm}^{const}$ denote the constant relationships from body coordinates to arm coordinates and from arm coordinates to forearm coordinates, respectively. ${}^{RG}R_{IMU0}^{init}$, ${}^{RG}R_{IMU1}^{init}$, and ${}^{RG}R_{IMU2}^{init}$ represent the rotation matrices of the three IMUs relative to the real global when initializing their positions, respectively.

After obtaining the constant matrix of the relevant bone, we calculate the updated orientation (rotation matrix) of the corresponding initial bone relative to the global coordinate system while maintaining a constant relationship during free motion.
(6)$$\begin{array}{cc} \; & {}^{RG}R_{Arm\_init}={}^{RG}R_{IMU0}\cdot {}^{Body}R_{Arm}^{const}\\ \; & {}^{RG}R_{Forearm\_init}={}^{RG}R_{IMU1}\cdot {}^{Arm}R_{Forearm}^{const} \end{array}$$
where ${}^{RG}R_{Arm\_init}$ and ${}^{RG}R_{Forearm\_init}$ are the corresponding initial bones, namely IMU1’ and IMU2’ in Figure 5, relative to the rotation matrix of the real global coordinates. ${}^{RG}R_{IMU0}$ and ${}^{RG}R_{IMU1}$ are the rotation relationship of the real global coordinates relative to the reference bone coordinates during motion.

Subsequently, during the operator’s movement, regardless of the human body movement or arm movement, the rotation matrix of the moving bone relative to the initial bone coordinate system is calculated as: (7)$$\begin{array}{c} \begin{array}{cc} \; & {}^{Arm\_init}R_{Arm}={\left({}^{RG}R_{Arm\_init}\right)}^{-1}\cdot {}^{RG}R_{IMU1}\\ \; & {}^{Forearm\_init}R_{Forearm}={\left({}^{RG}R_{Forearm\_init}\right)}^{-1}\cdot {}^{RG}R_{IMU2} \end{array} \end{array}$$
where ${}^{Arm\_init}R_{Arm}$ and ${}^{Forearm\_init}R_{Forearm}$ are the update directions of the arm and forearm relative to their respective initial bones during the movement and ${}^{RG}R_{IMU1}$ and ${}^{RG}R_{IMU2}$ are the rotation updates of the bones relative to the real global coordinate system during the movement.

Finally, the rotation matrix can be expressed as
(8)$$R=\left(\begin{array}{ccc} r_{11} & r_{12} & r_{13}\\ r_{21} & r_{22} & r_{23}\\ r_{31} & r_{32} & r_{33} \end{array}\right).$$
The Euler angle is calculated by the following formula, that is, the yaw angle ψ, the pitch angle η, and the roll angle *u*:(9)ψ=arctan(r21r21r11r11)η=arctan(r32r32r33r33)u=−arcsin(r31).

### 3.3. Sensor Fusion

(1)System modeling: considering the elbow’s pitch angle, we can express the linearized motion model for the pitch angle as follows:
(10)$$\begin{array}{cc} \; & \eta_{elbow,k}=\eta_{elbow,k-1}+T_{s}{\dot{\eta }}_{elbow,k}+\frac{T_{s}^{2}}{2}{\ddot{\eta }}_{elbow,k}\\ \; & {\dot{\eta }}_{elbow,k}={\dot{\eta }}_{elbow,k-1}+T_{s}{\ddot{\eta }}_{elbow,k} \end{array}$$
where $\eta_{elbow,k}$ and ${\dot{\eta }}_{elbow,k}$ represent the elbow pitch angle and its angular velocity at time *k*. $T_{s}$ represents the sampling time. $\eta_{elbow,k-1}$ and ${\dot{\eta }}_{elbow,k-1}$, respectively, denote the pitch angle and its angular velocity of the elbow joint at time k−1. ${\ddot{\eta }}_{elbow,k}$ represents acceleration and is regarded as a source of process disturbance. As a result, the state vector $x_{k}\epsilon R^{8}$ and the observation vector $y_{k}$ in Equation (Equation 11) are adopted for the complete system model.
(11)$$\begin{array}{cc} \; & x_{k}=\left[\psi_{shoulder,k},{\dot{\psi }}_{shoulder,k},\eta_{shoulder,k},{\dot{\eta }}_{shoulder,k},\right.\\ & {\left.u_{shoulder,k},{\dot{u}}_{shoulder,k},\eta_{elbow,k},{\dot{\eta }}_{elbow,k}\right]}^{T}\\ \; & y_{k}={\left[y_{shoulder\_w,k},y_{shoulder\_v,k},y_{shoulder\_u,k},y_{elbow\_v,k}\right]}^{T} \end{array}$$
Hence, the system model can be formulated as follows:
(12)$$\begin{array}{cc} \; & x_{k}=\Phi_{k}x_{k-1}+\Gamma_{k}w_{k}\\ \; & y_{k}=h_{k}x_{k}+v_{k} \end{array}$$
where $w_{k}={\left[{\ddot{\psi }}_{shoulder,k},{\ddot{\eta }}_{shoulder,k},{\ddot{u}}_{shoulder,k},{\ddot{\eta }}_{elbow,k}\right]}^{T}\;\epsilon \;R^{4}$, and $v_{k}$ are the process noise and the measurement noise, respectively. Their covariances are $Q_{k}$ and $R_{k}$, respectively. The matrices $\Phi_{k}\in R^{8\times 8}$ are state transition matrices, as shown in the following equation.
(13)$$\begin{array}{cc} \; & \Phi_{k}=diag\left(\phi_{k},\phi_{k},\phi_{k},\phi_{k}\right)\\ \; & \phi_{k}=\left[\begin{array}{cc} 1 & T_{s}\\ 0 & 1 \end{array}\right] \end{array}$$
And $\Gamma_{k}\in R^{8\times 4}$ is expressed as:
(14)$$\begin{array}{cc} \; & \Gamma_{k}=diag\left(\Sigma_{k},\Sigma_{k},\Sigma_{k},\Sigma_{k}\right)\\ \; & \Sigma_{k}=\left[\begin{array}{c} T_{s}^{2}/2\\ T_{s} \end{array}\right]. \end{array}$$(2)Kalman filter: The elbow pitch angle from Kinect and the IMU, the measurement vector, the observation matrix, and the covariance of the measurement noise are, respectively,
(15)$$\begin{array}{cc} \; & z_{k}={\left[y_{Kinect,k},y_{IMU,k}\right]}^{T}\in R^{8\times 1}\\ \; & H_{k}={\left[h_{Kinect,k},h_{IMU,k}\right]}^{T}\in R^{8\times 8}\\ \; & R=diag\left(R_{Kinect,k},R_{IMU,k}\right)\in R^{8\times 8}. \end{array}$$
The time update equation is
(16)$$\begin{array}{cc} \; & {\hat{x}}_{k|k-1}=\Phi_{k}{\hat{x}}_{k-1|k-1}\\ \; & P_{k|k-1}=\Phi_{k}P_{k-1|k-1}\Phi_{k}^{T}+\Gamma_{k}Q_{k}\Gamma_{k}^{T}. \end{array}$$
And the state update equation is
(17)$$\begin{array}{cc} \; & K_{k}=P_{k|k-1}H_{k}^{T}{\left(N_{k|k-1}\right)}^{-1}\\ \; & {\hat{x}}_{k|k}={\hat{x}}_{k|k-1}+K_{k}\delta_{k|k-1}\\ \; & P_{k|k}=\left(I-K_{k}H_{k}\right)P_{k|k-1}\\ \; & \delta_{k|k-1}=z_{k}-H_{k}{\hat{x}}_{k|k-1}\\ \; & N_{k|k-1}=H_{k}P_{k|k-1}H_{k}^{T}+R_{k}. \end{array}$$

### 3.4. Angle Algorithm for Data Glove

As shown in Figure 6, the data glove contains 6 inertial measurement units and 5 bending sensors. 6 inertial sensors are installed on the back of the hand and the fingertips of the five fingers, and 5 bending sensors are installed along the direction of finger extension. The specific parameters of the data gloves are shown in Table 2. The data fusion process uses the Kalman filter algorithm to perform in-depth data fusion of acceleration, angular velocity, magnetic value, and bending sensors to obtain high-precision attitude angles. The device communicates with the PC through the receiver, and the matching software fits the data of the sensor to calculate the three joint angles of each finger and sends it to a third-party device. We control the four fingers directly using the pitch angle of the first joint angle as the finger curvature. The pitch angle of the second joint is used for thumb bending, and the pitch angle of the first joint is used for thumb rotation and mapped to the rotation range of the manipulator. This comprehensive utilization of inertial measurement and bending sensor data enables us to capture the subtle movements of the hand and various gestures of the fingers very precisely, providing a strong support for the anthropomorphic control of the manipulator.

## 4. Experiment

### 4.1. IMU Angle Verification

According to the angle algorithm of the IMU system, the relative angle calculated by the two IMUs was compared with the angle from the wireless goniometer (W series, Biometrics Ltd., Ynysddu, UK) to verify the angle measurement accuracy of the IMU system. The wireless goniometer as shown in Figure 7 can measure the angle change in plane X and plane Y. We attached the two IMUs and the goniometer as shown in Figure 8 and stuck them together on the wrist. Angle changes in the X and Y planes of the two systems were measured by wrist flexion or extension and ulnar deviation. As shown in Figure 9, the experimental results emphasize the correctness of the angle measurement algorithm of the IMU system.

### 4.2. Angle Fusion between Kinect and IMU

The operator wore the IMU as shown in Figure 10 and paid attention to ensure that the y-axis of the IMU on the arm coincided with the axis of the bone. The operator stood within the effective range of the Kinect. In order to better control the robotic arm, the range of motion of the upper arm is as shown in Figure 11. We thus used the Kinect and IMU systems to simultaneously collect human motion gestures.

We set up four sets of actions to comprehensively verify the accuracy and stability of the fused data. Experimental maneuvers included (A) shoulder flexion, (B) shoulder abduction, (C) elbow flexion, and (D) shoulder rotation with 90° shoulder abduction. Each action collected shoulder yaw, shoulder pitch, shoulder roll, and elbow pitch for fusion comparison. The fusion curve was plotted, and the smoothness of the curve of single sensor and fused angles was compared. The trajectory smoothness of the curve under these four groups of actions was statistically calculated. The calculation formula is as follows. The smaller the data, the smoother the curve is. The results obtained are shown in Table 3.
(18)$$Smoothness=\sqrt{\frac{1}{n}\sum {\left(\ddot{\theta }\right)}^{2}}$$

The experimental results of action A are shown in Figure 12. Because the movement speed was too fast, the IMU data lagged behind the joint angle of Kinect, so it should not be too fast during the control process. The shoulder abduction angle collected by Kinect changed from 0 degrees to around 180 degrees during the collection period. This phenomenon is theoretically correct, but this mutation needs to be removed in control. In the position, the shoulder, elbow, and wrist are approximately in a straight line; thus, the shoulder roll calculated by Kinect is incorrect and is very different from that calculated by the IMU. More generally, the curve smoothness of joint angles after fusion is worse than that of Kinect, but it is not as smooth as the trajectory collected by the IMU for shoulder pitch and elbow pitch.

The experimental results of Action B are shown in Figure 13. Because this movement is often not standard, it results in accompanying changes in shoulder yaw angle and shoulder roll. More often than not, the elbow pitch angle is inaccurate. The smoothness of the fused trajectory is better than that of shoulder yaw and shoulder pitch, but it is worse for the two outer angles. Therefore, when controlling the robotic arm, the movements should be as standard as possible.

The experimental results of action C are shown in Figure 14. In this action, the position of the right arm moves within the spatial range composed of action A, action B and horizontal abduction action, which also limits the working space of the robotic arm to this quadrant, as shown in Figure 11b. There is an error in the initial position of the shoulder roll detected by Kinect. In addition, the joint angles under this action are well-detected, making the fused data more accurate. As shown in Table 3, the trajectory smoothness of the curves is significantly improved.

Action D’s experimental results are shown in Figure 15. In addition to the problem of Action B in this action, the shoulder pitch and elbow pitch measured by the IMU are wrong in the different action positions. This is also due to the irregularity of the action. However, The smoothness of the curve of this action has also been improved to some extent.

To sum up, the IMU can collect more subtle movements. Due to the high sensitivity of the IMU, it is able to sense shoulder rotation. The Kinect is more accurate at collecting shoulder roll when the elbow is flexed but cannot collect shoulder rotation when the shoulder, elbow and wrist are extended. The operator and the manipulator work in the first quadrant of the shoulder coordinate system. By fusing the Kinect and IMU data, we obtain more stable and accurate joint angle data. The fused data can make up for the shortcomings of various sensors alone and improve the robustness and reliability of the entire system. This is critical to achieving anthropomorphic control, as it requires robotic arms that can accurately mimic human movements, whether fine hand movements or overall upper body movements.

### 4.3. Data Glove Angle Verification

In this experiment, we aimed to evaluate the performance of an anthropomorphic control system in the control of fine hand movements. The action of grabbing a disposable paper cup filled with water and pouring water was designed, and the rotation angle of the wrist and thumb, as well as the bending angle of the metacarpophalangeal joints of the five fingers, were collected through the data glove. Both the rotation angle and bending angle of the thumb were mapped to [0, 90], which is more convenient to control the dexterous hand. The experimental results are shown in Figure 16. The wrist servo of the robotic arm can rotate up to 180 degrees, and the starting position was set to 0 degrees in the neutral position. Therefore, we set the wrist rotation angle value range to [−90, 90], which is easier to map to the robotic arm. Combined with the starting position of the servo, it is more suitable for the movement of the robotic arm. When the wrist pronated, the wrist rotation angle increased from 0 degrees to 90 degrees, which was mapped to the mechanical arm as pronation. When the wrist rotated externally, the angle changed from 0 to −90 degrees, and the robotic arm rotated externally. The stability of the data shows that the terminal control system has a high degree of operability and applicability in hand movement control.

### 4.4. Anthropomorphic Grip Control

This experiment aimed to verify the anthropomorphic control performance of a robotic system to perform complex pick and place tasks. Specifically, we aimed to control the movement of the robotic arm through the control system so that it could successfully grab a vertical mineral water bottle on the operating table and place it flat on the operating table. The robotic arm was mounted on a mobile chassis and lifting rod. The operator observed the distance between the robot arm and the operating table through his eyes. First, we let another operator control the chassis and lifting rod close to the console through the remote control so that the water bottle was within the working range of the mechanical right arm.

The experiment process is shown in Figure 17 and Figure 18. The starting positions of the operator and the robot are as shown in Figure 17a and Figure 18a. The operator first abducted the shoulder and controled the robot arm to reach the top of the operation platform, as shown in Figure 17b and Figure 18b. Second, the operator rotated the shoulder to control the robotic arm, as shown in Figure 17c and Figure 18c. After that, the operator bent the elbow to control the robotic arm to approach the water bottle, as shown in Figure 17d and Figure 18d. Again, the operator rotated the thumb to prepare for the grasping action, as shown in Figure 17e and Figure 18e. Then, the operator slightly flexed the shoulder and rotated to control the robot to reach the designated position, as shown in Figure 17f and Figure 18f. Again, the operator bent five fingers to control the dexterous hand to grab the water bottle, as shown in Figure 17g and Figure 18g. Then, the operator rotated the shoulder and wrist to control the mechanical arm to lift up and grab the water bottle horizontally, as shown in Figure 17h and Figure 18h. Again, the operator rotated the shoulder and spread five fingers to control the robot to place the water bottle on the operating table, as shown in Figure 17i,j and Figure 18i,j. Finally, the operator rotated the shoulder and then rotated the wrist and extended the elbow until the shoulder abducted the robot back to its initial position, as shown in Figure 17k,l and Figure 18k,l.

As shown in the Figure 19a, the trajectory of the robotic arm shows two drops in shoulder yaw and shoulder roll. The operator caught the water bottle around 40 s and dropped it around 60 s. The delay between the human and the robotic arm is approximately 0.05 s. Moreover, the fingertip tactile sensors of the manipulator can detect the maximum pressure on each finger, and the robot host receives pressure feedback. As shown in Figure 19b, after the finger was bent, the pressure value of the index finger reached more than $10^{4}\times 1/3000$ N. When the pressure exceeds a certain value, the manipulator cannot be controlled. When the pressure is lower than this value, the control is restored.

The experimental results show that our anthropomorphic control system can successfully complete the grasping task, not only reaching the target position stably but also grasping objects safely and precisely. The success of this experiment underscores the feasibility of our system in practical applications, especially when highly precise and complex operations are required, such as in industrial manufacturing and warehouse management. It also provides a solid basis for future research and applications, opening up new possibilities for the further development of robotics.

## 5. Discussion

Yi-Chun Du et al. proposed the use of an IMU to calibrate the joint angle data collected by Kinect, but only when the difference between the Kinect and IMU is large and limited to single-axis motion [41]. The Kalman filter we proposed fuses two sensors, which on the one hand makes the fusion result smoother and on the other hand is suitable for motion in the three axes of the shoulder joint. The fusion results were applied to a robotic arm, which greatly reduced the jitter of the robotic arm and improved the smoothness of the robotic arm’s movement. However, this method has high requirements for the standardization of movements. During control, the movements of the robotic arm should be adjusted appropriately when observing with eyes. More often than not, the range of motion of the robotic arm’s arm is limited to one quadrant.

Given the above limitations, we propose directions for future improvements. First, we can further investigate improved sensor fusion methods to address the accuracy of specific actions. Second, considering the sources of error in the different quadrants that the manipulator spans, we can adjust the location layout of the sensors or employ more complex fusion algorithms to improve the accuracy of the system. In addition, for high-speed movements or fast-changing situations, we can optimize the speed of data acquisition and the reaction time of the control system to ensure that the system can adapt more quickly to the needs of the operator. These improvements will help to improve the performance and applicability of the anthropomorphic control system.

Furthermore, to further improve the system, we plan to develop custom glove sensors in the future to reduce the cost and improve the scalability of the system. This self-developed glove sensor will be designed according to our specific needs, which can better meet the performance requirements of the anthropomorphic control system.

Human motion tracking systems can be developed to control various types of robots. For example, many researchers have developed soft robotic arms and graspers to perform flexible grasping tasks, and the motion tracking system developed in this article can be applied to the control of soft robotic arms and graspers [45,46]. A relationship model is established to convert the posture of the human body’s shoulders, elbows, and wrists into the target parameters of a multi-segment soft robotic arm to achieve an anthropomorphic posture. For adaptive grippers, the mobile structure of the gripper can be controlled by finger bending to open and close the gripper, making the gripping process more convenient. The human body’s remote operation has a very wide adaptability range and is more intuitive and easier.

## 6. Conclusions

In this study, we successfully developed an anthropomorphic control system with multi-sensor data fusion to achieve high-precision control of a robotic arm. The experimental results confirm the excellent performance and wide application prospects of the system. This research provides a valuable contribution to the fields of multisensor fusion and robot control. Future work will further explore the optimization of system performance and the expansion of application domains to meet the growing demands of automation and collaboration.

## Figures and Tables

**Figure 1 bioengineering-10-01243-f001:**
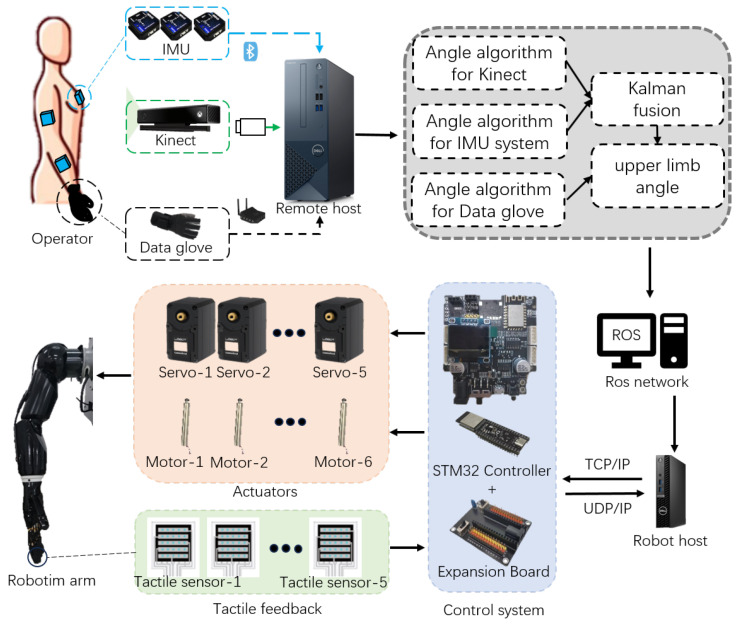
System structure.

**Figure 2 bioengineering-10-01243-f002:**
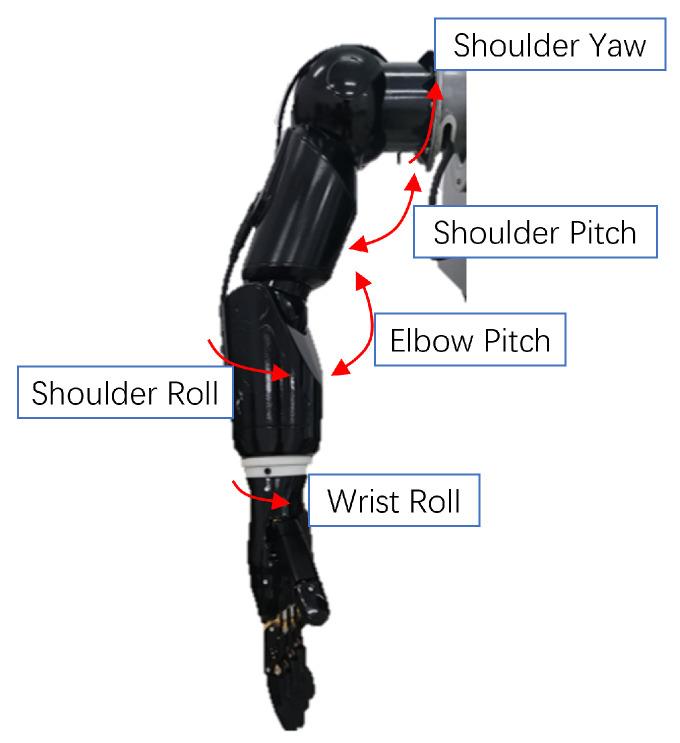
Demonstration of relative angles of the robotic arm.

**Figure 3 bioengineering-10-01243-f003:**
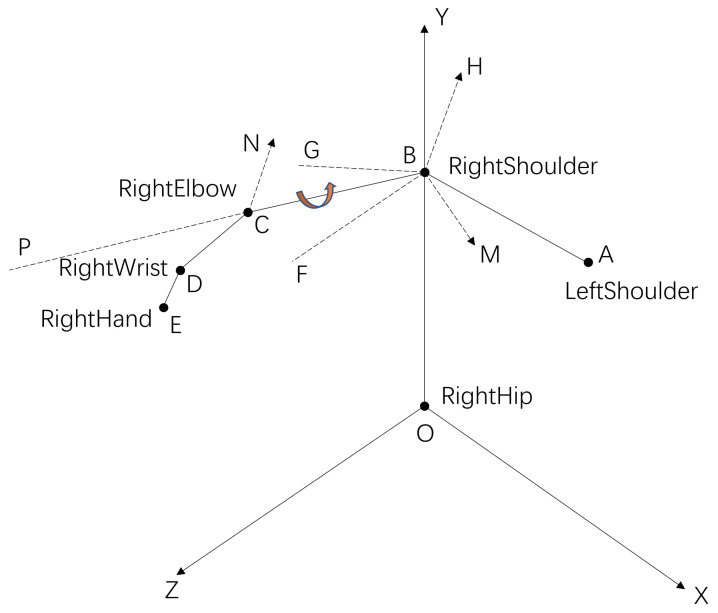
The method of calculating the joint angle from the bone points collected by Kinect.

**Figure 4 bioengineering-10-01243-f004:**
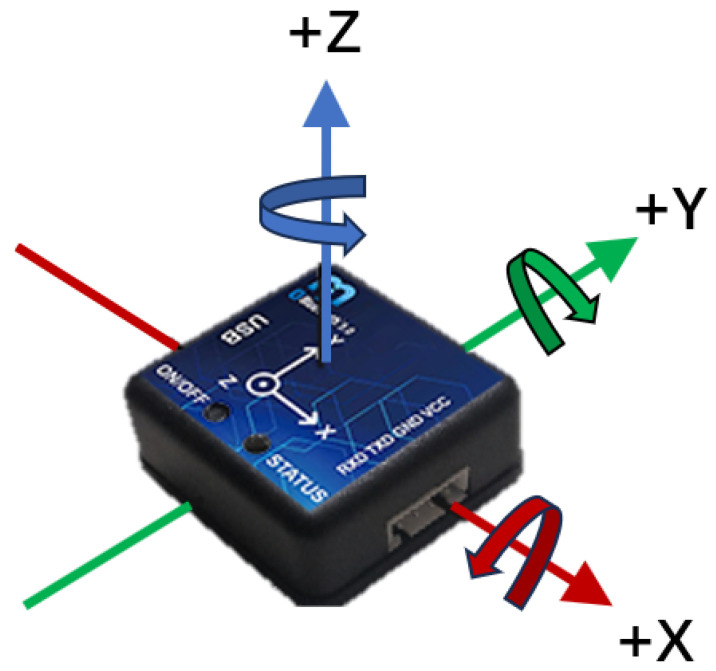
IMU axis.

**Figure 5 bioengineering-10-01243-f005:**
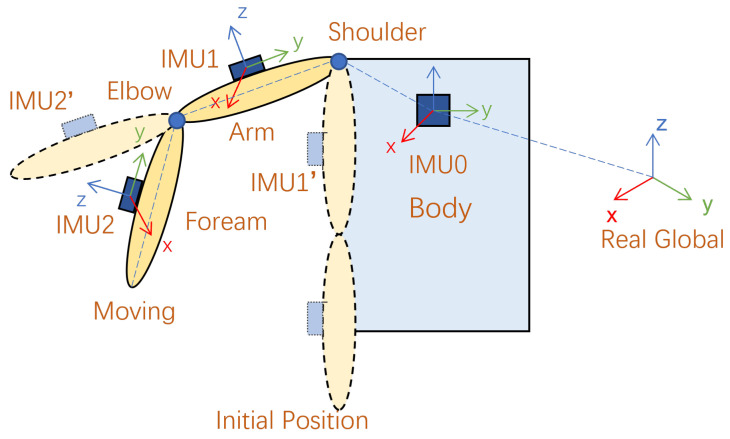
Kinematic model of the right upper limb.

**Figure 6 bioengineering-10-01243-f006:**
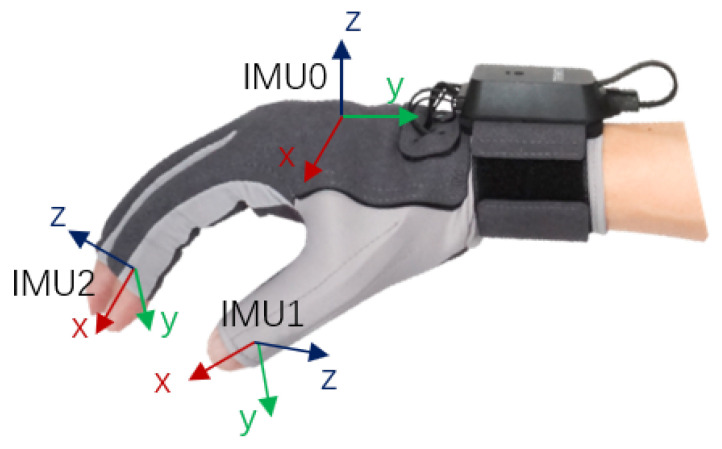
Data glove.

**Figure 7 bioengineering-10-01243-f007:**
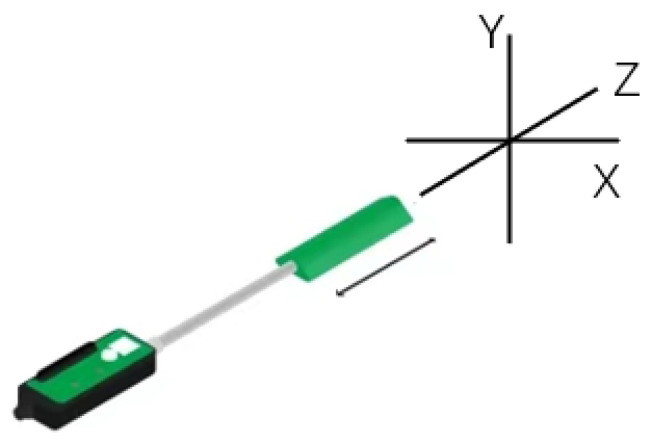
Wireless goniometer of Biometrics Ltd. (Ynysddu, UK).

**Figure 8 bioengineering-10-01243-f008:**
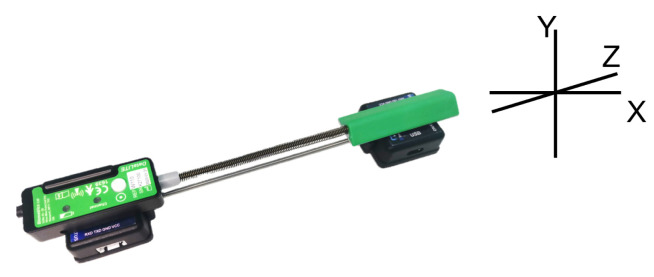
IMU system and goniometer.

**Figure 9 bioengineering-10-01243-f009:**
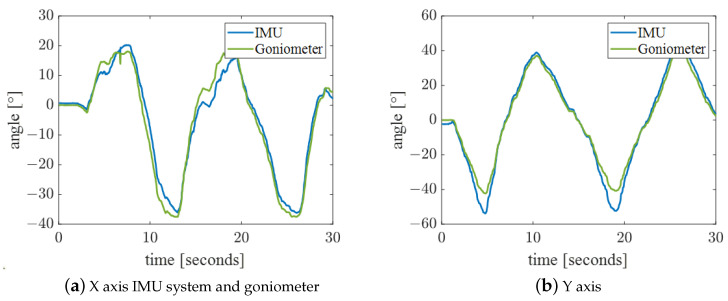
Comparison of IMU systems and goniometers for wrist flexion or extension and ulnar deviation.

**Figure 10 bioengineering-10-01243-f010:**
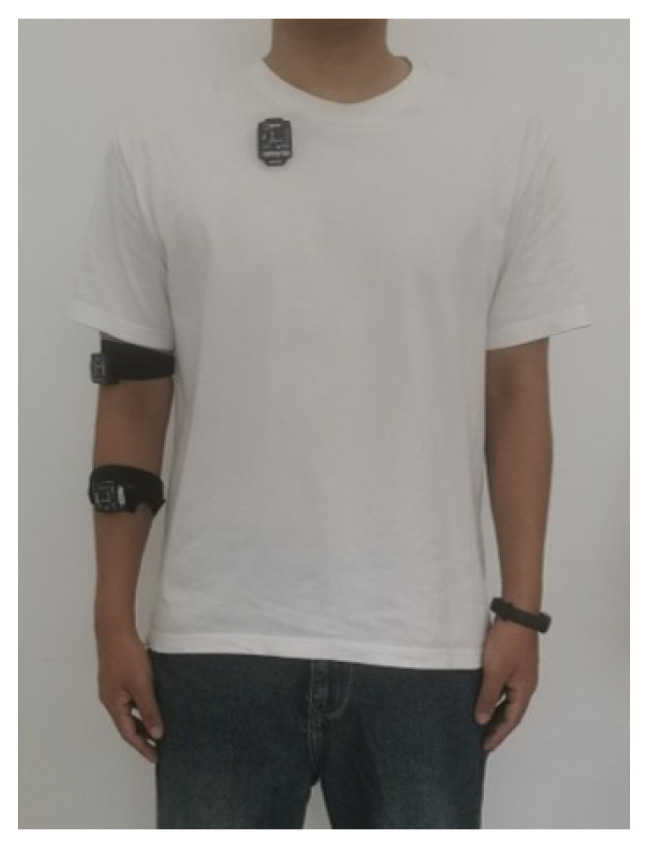
Wearing position of the IMU system.

**Figure 11 bioengineering-10-01243-f011:**
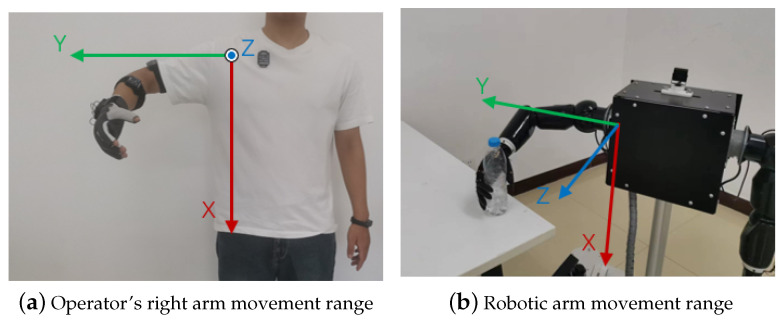
Movable quadrant of operation.

**Figure 12 bioengineering-10-01243-f012:**
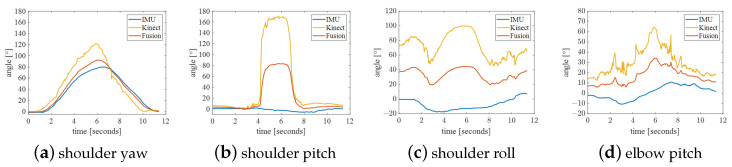
Action A: shoulder flexion.

**Figure 13 bioengineering-10-01243-f013:**
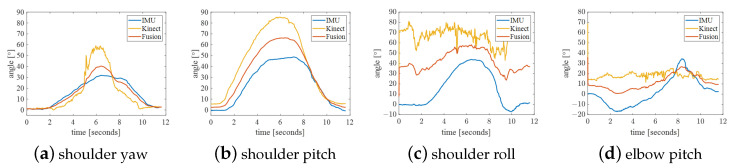
Action B: shoulder abduction.

**Figure 14 bioengineering-10-01243-f014:**
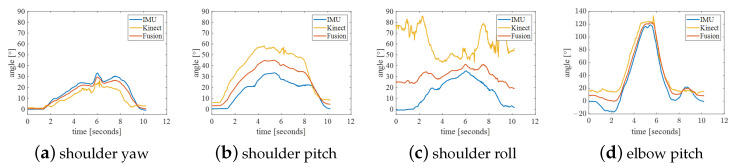
Action C: elbow flexion.

**Figure 15 bioengineering-10-01243-f015:**
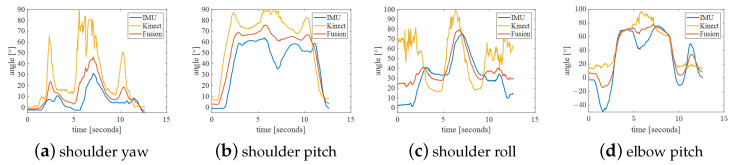
Action D: shoulder rotation with 90° shoulder abduction.

**Figure 16 bioengineering-10-01243-f016:**
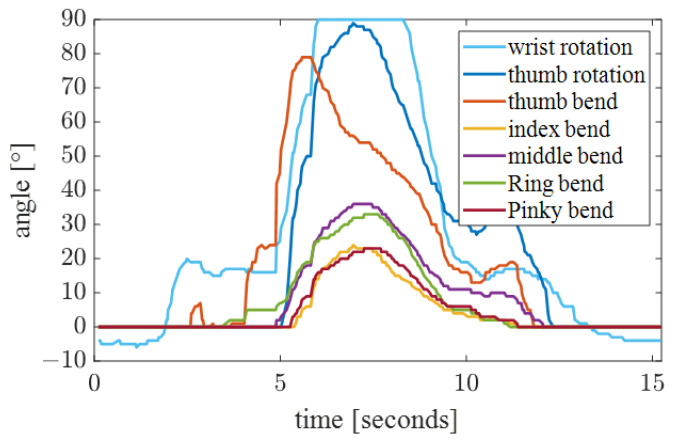
Data glove grip data.

**Figure 17 bioengineering-10-01243-f017:**
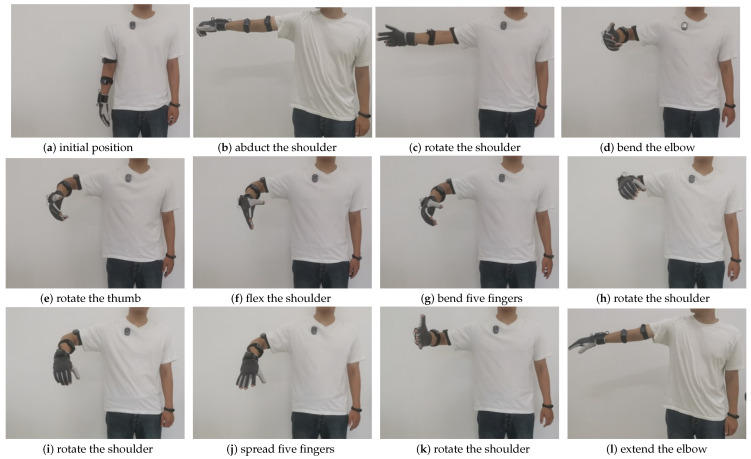
Operator movement group.

**Figure 18 bioengineering-10-01243-f018:**
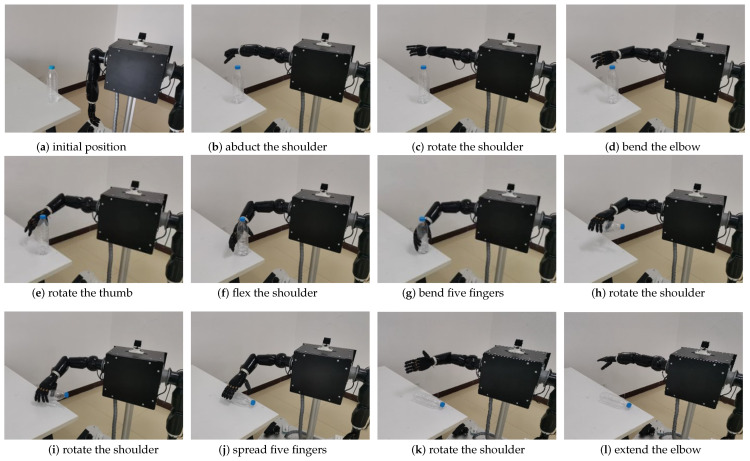
Robot anthropomorphic grasping.

**Figure 19 bioengineering-10-01243-f019:**
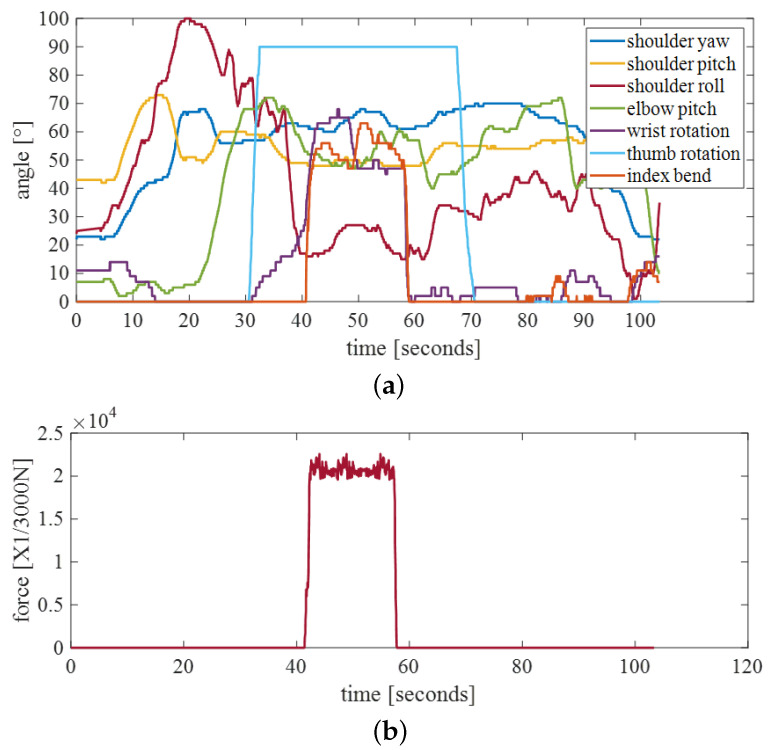
(**a**) Trajectory of robotic arm in griping experiment. (**b**) Index finger pressure in griping experiment.

**Table 1 bioengineering-10-01243-t001:** Single IMU parameter table.

Parameter Name	Value
Acceleration range	±16 g (resolution 0.00048 g)
Angular velocity range	±2000 deg/s (resolution 0.061 deg/s)
magnetic field range	±8 G
Acceleration accuracy	0.01 g (i.e., 0.098 m/s^2^)
Angular velocity accuracy	0.06 deg/s
Static Euler angle accuracy	X and Y angles: 0.05 deg, Z angle: 0.1 deg
Dynamic accuracy	0.5 deg
Reported frame rate	0.5–250 Hz (adjustable)

**Table 2 bioengineering-10-01243-t002:** Data glove parameter table.

Parameter Name	Value
Dynamic accuracy	Roll/Pitch ≤ 1 deg Pitch ≤ 2deg (RMS)
Static accuracy	Roll/Pitch ≤ 0.2 deg Pitch ≤ 1deg (RMS)
Acceleration measurement range	±16 g
Angular velocity range	±2000 dps
Angle measurement resolution	0.02 deg
Maximum rate	100 fps
Wireless transmission frequency band	2.4 GHz/5.8 GHz

**Table 3 bioengineering-10-01243-t003:** Curve smoothness results.

Movement	Shoulder Yaw			Shoulder Pitch			Shoulder Roll			Elbow Pitch		
	IMU	Kinect	Fusion	IMU	Kinect	Fusion	IMU	Kinect	Fusion	IMU	Kinect	Fusion
action A	260.48	1416.27	186.69	163.16	2703.29	331.63	167.78	2235.09	158.09	204.69	3336.45	286.77
action B	133.94	2048.15	146.87	231.34	567.11	117.39	452.14	5247.6	936.47	375.94	3380.8	1225.9
action C	235.02	958.3	161.41	321.96	794.61	169.35	457.3	3389.55	355.55	798.1	2009.91	420.08
action D	335.7	4465.34	333.73	474.56	939.98	249.14	645.22	5399.79	527.62	852.05	2138.36	468.42

## Data Availability

Not applicable.

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
