# Peer review of "Sensor Fusion-Based Anthropomorphic Control of a Robotic Arm"

_bioengineering, 2023, doi:10.3390/bioengineering10111243_

Round 1

Reviewer 1 Report

This paper describes mapping of motion capture data to robotic arm control signal. The methodology sounds. I have following minor comments:

Abstract: "In order to improve the accuracy and stability of the data" - in my opinion it is better to write "motion tracking" instead of "data".

Please write more details about hardware you use, especially glove and IMU sensor. Please list scale range, resolution, static and dynamic drift.

Page 5: "This IMU system is equipped with a self-developed attitude calculation algorithm, which has many advantages, such as strong real-time performance, high precision, good stability and no drift." The drift is among most important problem in IMU-based motion tracking system. Please give more details how this algorithm works. What does it mean "self-developed" - does it mean, that author of this paper developed that algorithm?

Equation (9) - authors work on Euler angles, what about the gimbal lock? Why authors do not use quaternions?

Figure 17 – what about situation when wrist rotation > 90? Why there are no values > 90 on the plot?

Reviewer 2 Report

The paper presents a sensor fusion approach for controlling an antropomorphic robotic manipulator. The topic of the paper is interesting. The manuscript is overall clear and easy to follow. However, the following points need to be clarified in order to improve the quality of the paper.

1) The main advantages and disadvantages of the proposed approach with respect to the other similar methods available in the present literature should be clearly described and commented. Furthermore, it is not clear what is the novelty in the mathematical formulation reported in the paper.

2) The overall software architecture should be clearly described in the text. The reader should be able to reproduce the experimental results.

3) The trajectories of the robot should be reported in the manuscript, e.g., joint positions over time.

4) The overall quality of the figures is low. I suggest increasing the quality of the figures using vector graphics images.

5) Differences between "IMU", "Kinect" and "Fusion" approaches should be quantified using quantitative metrics (root mean square error, absolute error, etc.). What is the estimated delay between human motion and robot motion?

6) The literature review should be extended, by including additional options for the remote control of a manipulator. See for instance:

https://doi.org/10.3390/robotics10020054

https://ieeexplore.ieee.org/document/9635879

Reviewer 3 Report

This paper presented an anthropomorphic control system that utilizes multiple sensor technologies to achieve precise control of a robotic arm. Overall, the paper is well written and has solid contribution. However, several important issues should still be addressed. Below are some comments for the authors to consider:

1. In the experiment section 4.4, the proposed control system was used to control a previously developed robot arm to perform complex pick and place tasks. How do the authors ensure that the water bottle in Figure 19 was well grasped and not squeezed? Does the proposed system also have force sensors to measure the grasping force?

2. In the experiment, how do the operator ensure that the robot arm reaches the operation platform? Does the operator use eyes or cameras to determine the spatial distance between the robot arm and the operation platform?

3. In the current state of the art, many researchers have developed soft robotic arms or grippers (see the related work below) to perform flexible grasping tasks. From this perspective, the authors are also recommended to mention those work in the Discussion section (Section 5) and discuss the feasibility of the proposed sensor-fusion-based control system for those soft robot arms or grippers. Below are several related work for soft robotic arm or gripper:

"LARG: A Lightweight Robotic Gripper With 3-D Topology Optimized Adaptive Fingers". https://doi.org/10.1109/TMECH.2022.3170800

"Design and Implementation of a Soft Robotic Arm Driven by SMA Coils". https://doi.org/10.1109/TIE.2018.2872005

Round 2

Reviewer 3 Report

The authors have revised the manuscript according to my comments. Therefore, I recommend to publish this paper in the journal.
